# Quality-adjusted life year comparison at medium term follow-up of endovascular versus open surgical repair for abdominal aortic aneurysm in young patients

**Eunae Byun** [1], **Tae-Won Kwon** [2]*, **Hyangkyoung Kim**[2], **Yong Pil Cho**[2], **Youngjin Han**[2], **Gi Young Ko**[3], **Min-Jae Jeong**[4]

**1** Department of Surgery, Dona-A University Hospital, Seo-gu, Busan, Republic of Korea, **2** Division of Vascular Surgery, Department of Surgery, Asan Medical Center, University of Ulsan College of Medicine, Songpa-gu, Seoul, Republic of Korea, **3** Division of Interventional Radiology, Department of Radiology, Asan Medical Center, University of Ulsan College of Medicine, Songpa-gu, Seoul, Republic of Korea, **4** Department of Surgery, Gangneung Asan Hospital, University of Ulsan College of Medicine, Sacheon-myeon, Gangneung, Republic of Korea

\* twkwon2@amc.seoul.kr

## Abstract

### Objectives

This study aimed to compare the quality of life and cost effectiveness between endovascular aneurysm repair (EVAR) and open surgical repair (OSR) in young patients with abdominal aortic aneurysm (AAA).

### Design

This was a single-center, observational, and retrospective study.

### Materials and methods

A retrospective analysis was conducted of patients with AAA, who were <70 years old and underwent EVAR or OSR between January 2012 and October 2016. Only patients with aortic morphology that was suitable for EVAR were enrolled. Data on the complication rates, medical expenses, and expected quality-adjusted life years (QALYs) were collected, and the cost per QALY at three years was compared.

### Results

Among 90 patients with aortic morphology who were eligible for EVAR, 37 and 53 patients underwent EVAR and OSR, respectively. No significant differences were observed in perioperative cardiovascular events and death between the two groups. However, during the follow-up period, patients undergoing OSR showed a significantly lower complication rate (hazard ratio [HR] = 0.11; P = .021). From the three-year cost-effectiveness analysis, the total sum of costs was significantly lower in the OSR group (P < .001) than that in the EVAR group, and the number of QALYs was superior in the OSR group (P = .013). The cost per

**Data Availability Statement:** All relevant data are within the manuscript and its Supporting Information files.

**Funding:** The authors received no specific funding for this work.

**Competing interests:** The authors have declared that no competing interests exist.

QALY at three years was significantly lower in the OSR group than that in the EVAR group (mean: $4038 vs. $10 137; respectively; P < .001)

## Conclusions

OSR had lower complication rates and better cost-effectiveness than EVAR Among young patients with feasible aortic anatomy.

## Introduction

Currently, endovascular aneurysm repair (EVAR) and open surgical repair (OSR) are the only options for treating abdominal aortic aneurysm (AAA). Previous randomized controlled trials (RCTs) that compared the outcomes of EVAR and OSR have reported perioperative, 30-day, and short-term outcomes of EVAR to be superior than those of OSR, but other trials reported OSR to be a safer option than EVAR in terms of long-term outcomes [1–4]. The unfavorable long-term outcomes of EVAR are owing to the various complications and situations that eventually require reintervention. The possibility of an endoleak increases over time with morphological changes in the aorta. Even type 2 endoleaks (T2ELs), which are considered benign, have the potential to expedite the occurrence of other types of endoleaks, if presented with sac expansion [5].

EVAR and OSR have their respective advantages and disadvantages, which are elaborated in the National Institute for Health and Care Excellence guidelines [6]. This set of guidelines recommends OSR as the standard treatment for AAA. It emphasizes that if patients with AAA choose to undergo EVAR, the clinicians should ensure that the patients understand the potential complications and the possibility of secondary intervention that are associated with EVAR [6].

EVAR is the preferable choice for older patients with comorbidities because of its desirable periprocedural outcomes [1–4]. However, in younger patients who have longer life expectancies and for whom age is not a factor, the selection of treatment is less obvious, for the aforementioned reasons. This dilemma becomes more prominent when the aortic morphology of young patients makes them eligible for both EVAR and OSR. In these cases, health care professionals need to consider the health-related quality of life (HRQOL) and the cost of each surgical method when making the decision.

This study compared the quality-adjusted life years (QALYs) and medical expenses and reported the complication rates, including the reinterventions associated with increased cost, in young patients (<70 years old) with an aortic anatomy that is eligible for both EVAR and OSR. The results of this study aim to offer a foundation for the decision-making process of whether to perform an elective EVAR or OSR in young patients with AAA.

## Methods

### Study design and patient selection

We conducted a retrospective analysis on a prospectively compiled and computerized database of consecutive patients with AAA. All patients under 70 years of age who underwent elective EVAR or OSR for AAA between January 2012 and October 2016 and had their follow-ups terminated on March 30, 2020, were included. The indication for AAA repair was based

primarily on the maximum diameter of the aneurysm, which was at least 5.5 cm for males and 5.0 cm for females [7].

All patients in the present study underwent preoperative contrast-enhanced computed tomography (CT). Following the analysis of the CT images by a professional radiologist at the hospital, one of the authors analyzed whether the aortic morphology conformed to a composite list of instructions for the inclusion of only patients with the aortic morphology that was suitable for EVAR and OSR: aneurysm infrarenal neck length ≥10 mm, infrarenal aortic neck angulation <60 degrees, neck diameter between 18 and 28 mm, and common iliac artery distal fixation length ≥10 mm [8]. Patients meeting one or more of the following exclusion criteria were not eligible to participate in this study 1) patients with aortic morphology that is not feasible for EVAR; 2) patients requiring emergency surgery due to a rupture or infection and requiring only OSR owing to the inflammation (only degenerative AAA were included); 3) patients who underwent other types of surgery for a different illness (e.g., cancer) with AAA treatment; 4) foreign patients (because of a different cost system); and 5) patients at high risks from general anesthesia (GA) based on cardiac, pulmonary, and neurologic examination, because they could only undergo EVAR under local anesthesia (LA) (Fig 1) were excluded. The possibility of general anesthesia was evaluated by collecting the opinions of experts in each department, and patients evaluated as high-risk groups during GA were excluded. Even when EVAR was always performed under LA, we prepared for GA, to be possible just in case.

The choice between EVAR and OSR was based on patient preference. After preoperative evaluation, the patients made a decision after consulting with the doctor in charge (vascular surgeon and radiologist) about the treatment options [9]. When performed under LA, the patients had the advantage of reducing the risk from GA. However, patients had to lie on a firm bed during the long procedures and became anxious. The patients carefully considered the strengths and weaknesses of each treatment and chose accordingly.

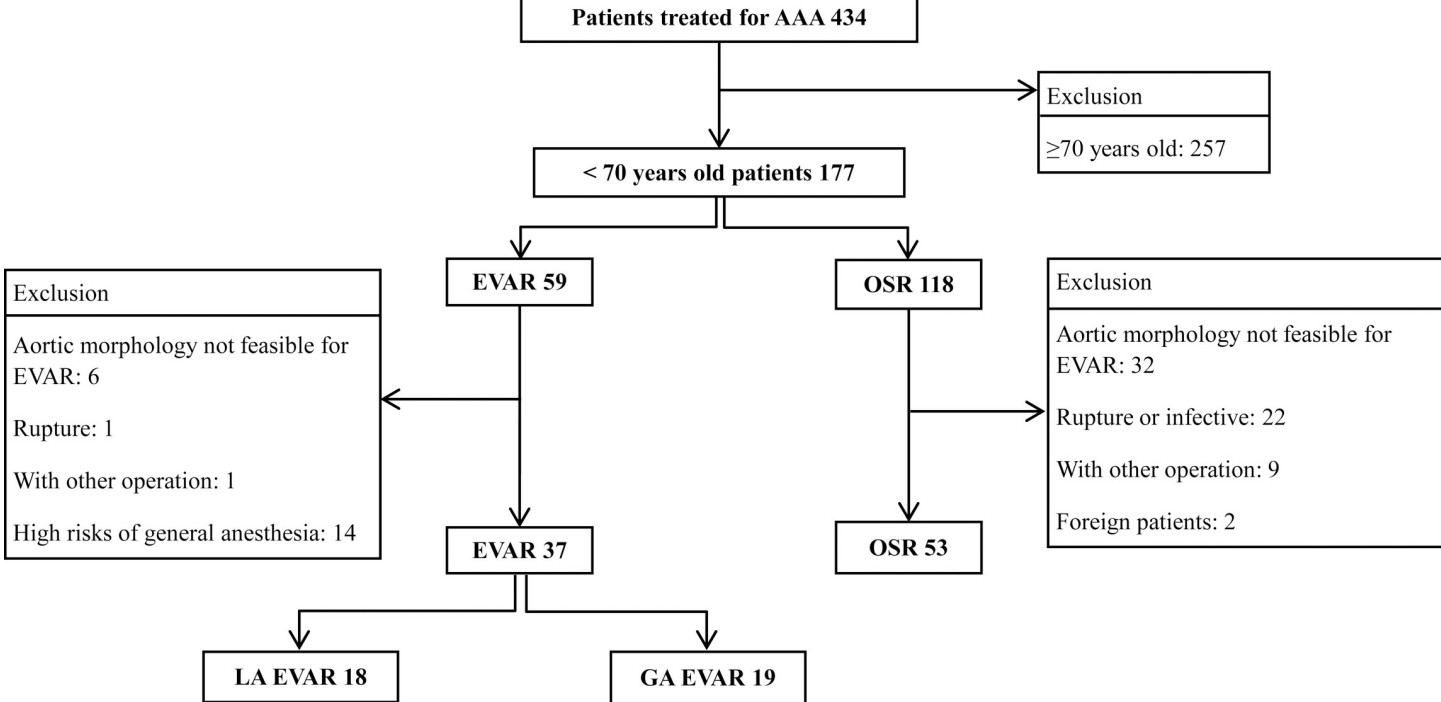

**Fig 1. Enrollment of the study.** EVAR, Endovascular aneurysm repair; OSR, open surgical repair; LA, local anesthesia; GA general anesthesia.

This study was approved by the institutional Review Board of Asan Medical Center (2020–0215). The requirement for informed consent was waived owing to the retrospective nature of this study.

## Procedure details

All OSRs were performed by two vascular surgeons. There was no difference in following a single protocol in the same hospital setting. EVARs were performed by one vascular surgeon or two radiologists; they had often operated together and discussed all results together. Among the EVAR procedures, 48.6% (18/37) were performed by radiologists and a percutaneous approach was used under inguinal LA in an intervention room. The remaining EVAR (19/37; 51.4%) procedures were performed by surgeons in operating rooms using bilateral surgical cutdown under GA.

The OSR procedures were performed using a transperitoneal approach with an incision from the xiphoid process to the pubis. The standard procedure involved infrarenal clamping and reconstruction by performing interposition with a bifurcated graft. Gortex (W. L. Gore & Association, Flagstaff, AZ, USA) and Dacron (Hemagard, Maquet Getinge Group, Rastatt Germany) grafts were used in 12 (22.6%) and 41 cases (77.4%), respectively.

For EVAR, different commercially available stent-graft devices were used. Endurant (Medtronic Inc., Santa Rosa, CA, USA) devices were used in most cases (28/37; 75.7%), while Zenith (Cook Medical, Bloomington, IN, USA) and Excluder (W. L. Gore & Associations, Flagstaff, AZ, USA) devices were used in eight (21.6%) and one (2.7%) patients, respectively. Hemostasis was achieved using Perclose Prostar XL or Proglide devices (Abbott Vascular, Redwood City, CA, USA) in all patients undergoing EVAR under LA.

## Follow-up

The frequency of outpatient visits and follow-up tests were based on an expert's opinion. The follow-up protocol included a physical examination and CT imaging. The patients who underwent EVAR visited the hospital regularly at 30 days, 6 months, 1 year, and every 2 years thereafter. Enhanced CT (aortic dissection CT or lower extremity CT angiography) was performed before visiting the outpatient department. In contrast, the patients who underwent OSR visited the outpatient department for medication every 6 months. CT was conducted only for patients who presented with other aortic aneurysms, abnormal symptoms, or abnormal findings on physical examination.

## Outcomes

The primary outcomes that were analyzed included medical expenses, QALYs, and cost per QALY at three years. The secondary outcomes considered were readmission-requiring complication rates, including reintervention that resulted in increased cost and change in QALY, all-cause death, and aneurysm-related death in the perioperative period (30 days) and during follow-up.

The medical expenses comprised costs for in-hospital care, routine follow-ups, and complications. The data for the in-hospital and complication-related costs was extracted from the records of the patients' total medical expenses that were not adjusted for health care benefits. The total sum of costs included the doctor's fees; costs of equipment for operations or interventions, patient rooms, imaging examinations, and laboratory examinations; and other costs. On the other hand, data on the follow-up costs that were extracted from the records of unit costs of a routine aortic dissection CT ($393 in 2012) were based on the costs in our center. The price was based on the year of 2012 (the year in which the recruitment for this study started),

and all costs were collected in South Korean won. The costs in future years were discounted at the rates of 1.3%, 2.6%, 3.3%, 4.3%, 6.3%, 7.9%, 8.3%, and 10.0% per year from 2013 to 2020 (according to the Korean Statistical Information Service consumer price index) [10]. Subsequently, all costs were expressed in US dollar (USD, $) at the exchange rate prevalent in 2012 [11].

We used the QALYs to calculate the HRQOL. The HRQOL values are recommended to be calibrated using social preference weights elicited from the general population. Thus, we used quality weights from the Euro Quality of Life–5 Dimensions (EQ-5D) records. We used the average score of EQ-5D during the first three years after the primary procedure by contacting and enquiring the patients through the phone during the investigation period. The following formula indicates the South Korean population-based preference weights for the EQ-5D [12–15]: $Y = 1 - (0.05 + 0.096 \times M2 + 0.418 \times M3 + 0.046 \times SC2 + 0.136 \times SC3 + 0.51 \text{ v } UA2 + 0.208 \times UA3 + 0.037 \times PD2 + 0.151 \times PD3 + 0.043 \times AD2 + 0.158 \times AD3 + 0.05 \times N3)$, where M2, mobility level 2; M3, mobility level 3; SC2, self-care level 2; SC3, self-care level 3; UA2, usual activity level 2; UA3, usual activity level 3; PD2, pain or discomfort level 2; PD3, pain or discomfort level 3; AD2, anxiety or depression level 2; AD3, anxiety or depression level 3; and N3, any dimension on level 3.

## Statistical analysis

Categorical variables were compared using the Pearson chi-square tests or Fisher's exact test, as appropriate. Continuous variables were analyzed using the Student's t-test and Wilcoxon Mann–Whitney *U* test after the normality test. Costs were presented using the mean values rather than the median values, which is considered appropriate in health economic evaluations.

For comparison of the complications, the possible differences between the groups were tested using recurrent event models (Andersen and Gill model with a robust sandwich estimator). Hazard ratio (HR) was adjusted for age because the age of the OSR group was significantly lower according to the analysis of the patients' basic characteristics.

The cardiovascular events and death rates were calculated by the Kaplan–Meier survival analysis and were compared using the log-rank test.

$P < .05$ was considered as statistically significant. All statistical analyses were performed using IBM SPSS Statistics for Windows, version 21.0 (IBM Corp, Armonk, NY, USA).

## Results

### Patient characteristics

A total of 434 patients underwent EVAR or OSR for AAA between January 2012 and October 2016. Among these, 257 patients were excluded from the present study as they were over the age of 70 years. From the remaining patients, 59 and 118 patients underwent EVAR and OSR, respectively. After applying the exclusion criteria of patients with aortic morphology feasible for only EVAR and those at high risk from GA, we finally included 90 patients in this study. Of these 90 patients, 37 (41.1%) had undergone an EVAR and 53 (58.9%) had undergone an OSR. (Fig 1) A total of 53 patients, who had undergone an OSR, had aortic morphology that was feasible for an EVAR but had chosen to undergo an OSR.

In this study, 83 of 90 patients were male (92.2%), and the median age was 64.0 years (interquartile range, 60.8–66.3 years). The median age in the EVAR group was higher than that in the OSR group (66.0 vs. 63.0 years, respectively; $P < .001$); however, other demographic factors and comorbidities did not significantly differ between the groups. Preoperative CT showed a mean aortic maximum diameter of 58.6 ± 10.8 mm; 57.8 ± 9.5 mm in the EVAR group and

**Table 1. Baseline demographics and aortic aneurysm morphological characteristics.**

| | Total (n = 90) | EVAR (n = 37) | OSR (n = 53) | P-value[†] |
|---|---|---|---|---|
| **Baseline characteristics** | | | | |
| **Age (years)** | 64.0 (60.8–66.3) | 66.0 (63.0–68.0) | 63.0 (58.5–65.0) | < .001[‡] |
| **Male sex** | 83 (92.2) | 34 (91.9) | 49 (92.5) | >.05[§] |
| **BMI (kg/m²)** | 24.9 (± 3.0) | 25.1 (± 3.4) | 24.8(± 2.7) | .632[‡] |
| **Smoking** | 37 (41.1) | 14 (37.8) | 23 (43.4) | .598 |
| **Hypertension** | 48 (53.3) | 16 (43.2) | 32 (60.4) | .109 |
| **DM** | 16 (17.8) | 5 (13.5) | 11 (20.8) | .377 |
| **COPD** | 8 (8.9) | 4 (10.8) | 4 (7.5) | .712[§] |
| **CKD** | 15 (16.7) | 8 (21.6) | 7 (13.2) | .292 |
| **PAOD** | 1 (1.1) | 0 (0) | 1 (1.9) | >.05[§] |
| **CAD** | 30 (33.3) | 10 (27.0) | 20 (37.7) | .289 |
| **HF** | 2 (2.2) | 1 (2.7) | 1 (1.9) | >.05[§] |
| **Cancer Hx.** | 9 (10.0) | 6 (16.2) | 3 (5.7) | .153[§] |
| **CVA** | 7 (7.8) | 3 (8.1) | 4 (7.5) | >.05[§] |
| *CCI* | 2.0 (1.0–3.0) | 2.0 (1.0–3.0) | 2.0 (1.0–3.0) | .795[*] |
| **Aortic aneurysmal morphology** | | | | |
| **Maximum aortic diameter (mm)** | 58.6 (± 10.8) | 57.8 (± 9.5) | 59.2 (± 11.6) | .556[‡] |
| **Aortic neck length (mm)** | | 36.3 (± 12.1) | 35.4 (± 15.2) | .761[‡] |
| **Aortic neck width (mm)** | | 20.6 (± 2.5) | 20.1 (± 1.9) | .404[‡] |
| **Infrarenal aortic angulation (˚)** | | 31.7 (± 25.4) | 36.8 (± 28.5) | .385[‡] |
| **Shorter CIA length (mm)** | | 35.9 (± 14.8) | 33.0 (± 12.5) | .084[‡] |

Values in parentheses are percentages, unless age, BMI, CCI, and aortic aneurysmal morphology; age and CCI are reported as median (interquartile range); BMI and aortic aneurysmal morphology values are reported as means (± standard deviation).

†Pearson chi-square test, except.

‡Student's t-test.

§Fisher's exact test.

* Mann-Whitney *U* test.

EVAR, Endovascular aneurysm repair; OSR, open surgical repair; BMI, body mass index; DM, diabetes mellitus; COPD, chronic obstructive pulmonary disease; CKD, chronic kidney disease; PAOD, peripheral arterial occlusive disease CAD, coronary artery disease; HF, heart failure; Cancer Hx., cancer history; CVA, cerebrovascular accident; CCI, Charlson Comorbidity Index; CIA, common iliac artery.

59.2 ± 11.6 mm in the OSR group (P = .556; a difference that was not statistically significant). Other anatomical details also showed no significant differences between the two groups (Table 1).

## Follow-up results

The median follow-up duration was 52 months (interquartile range, 42.8–70.5 months); 49 months (interquartile range, 41.5–73.0 months) in the EVAR group and 55 months (interquartile range, 44.0–69.5 months) in the OSR group. Perioperative myocardial infarction (MI) occurred only in the OSR group (n = 1; P = .403), but no perioperative cerebrovascular accidents (CVAs) or deaths (all-cause or aneurysm-related) occurred in either group. The differences in the rates of late MI, CVA, and death (all-cause, aneurysm-related) between EVAR and OSR were negligible (MI, P = .095; CVA, P = .403; aneurysm-related death, P = .229; all-cause death, P = .153). One aneurysm-related death (AAA rupture) occurred in the EVAR group 31 months after the procedure (Table 2). Fig 2 shows the survival curves. There was no statistically significant difference in all-cause death (P = .153) and aneurysm-related death

**Table 2. Cardiovascular events and death rates in perioperative and during the follow-up period.**

| | Perioperative outcomes | | | Follow-up outcomes | | |
|---|---|---|---|---|---|---|
| | EVAR (n = 37) | OSR (n = 53) | P-value[†] | EVAR (n = 37) | OSR (n = 53) | P-value[†] |
| MI | 0 (0) | 1 (1.9) | .403 | 6 (16.2) | 3 (5.7) | .095 |
| CVA | 0 (0) | 0 (0) | NA | 0 (0) | 1 (1.9) | .403 |
| Aneurysm-related death | 0 | 0 | NA | 1 (2.7) | 0 (0) | .229 |
| All-cause death | 0 | 0 | NA | 6 (16.2) | 4 (7.5) | .153 |

Values in parentheses are percentages.

†Log-rank test.

EVAR, Endovascular aneurysm repair; OSR, open surgical repair; MI, myocardial infarction; CVA, cerebrovascular accident; NA, not applicable.

(P = .229) between the EVAR and OSR groups, but there was a significant difference in complication (requiring admission)-free survival (P = .020).

At the median follow-up time of 52 months (interquartile range, 42.8–70.5 months), reinterventions had been performed in 13 patients, among which eight patients had undergone EVAR (21.6%). Readmission-requiring complications occurred in three cases in three patients who had undergone OSR (5.7%); the difference was statistically significant between EVAR and OSR (HR, 0.11; 90% confidence interval [CI], 0.03–0.45; P = .0022; Table 3). This result was adjusted for age as this factor differed significantly between the two groups (P < .001; Table 1). One patient in the EVAR group required three reinterventions, and three patients required two reinterventions each. In detail, of the 13 reinterventions performed in the EVAR group, six involved embolization due to T2EL and type 1a endoleak (T1aEL). A stent graft insertion due to type 1b endoleak (T1bEL), T2EL, and limb occlusion were observed in four cases. Open conversions due to T2EL or endotension were observed in two cases; open conversions were included in the reintervention count. In the OSR group, there were three cases of complications. One case involved conservative care under readmission due to mechanical ileus and two cases involved herniorrhaphies due to incisional hernias. The three cases were considered as first complications (Table 4).

Both the EVAR and OSR groups included patients with complications that required no readmission or reintervention. For example, in the six patients from the EVAR group, T2ELs were found that either disappeared on their own or lacked sac growth. In the OSR group, three patients were diagnosed with retrograde ejaculation, and received symptomatic treatment. Complications, such as graft infection, distal embolization, and renal infarction, were not found in this study.

## Cost analysis

The costs of hospitalization for the primary procedure, follow-up, and complications were accumulated and calculated annually. As the year passed, the difference between EVAR and OSR increased. It was found that not only the in-hospital cost but also the cumulative cost at one, two, and three years was statistically significantly higher in EVAR compared with that of OSR. Our three-year HRQOL analysis showed that the QALY at three years was significantly higher in the OSR group (mean, 2.69; 95% CI, 2.69–2.82) compared with that of the EVAR group (mean, 2.49; 95% CI, 2.27–2.70) (P = .013). Overall, the cost per QALY at three years was significantly lower in the OSR group (mean, $4038; range, $2705–$12 545) than that of the EVAR group (mean, $10 137; range, $4454–$68 419) (P < .001; Table 5).

In our three-year cost analysis, the in-hospital cost for EVAR (mean, $16 498; range, $10 905–$34 296) vs. OSR (mean, $10 523; range, $7615–$25 105), and the follow-up costs for

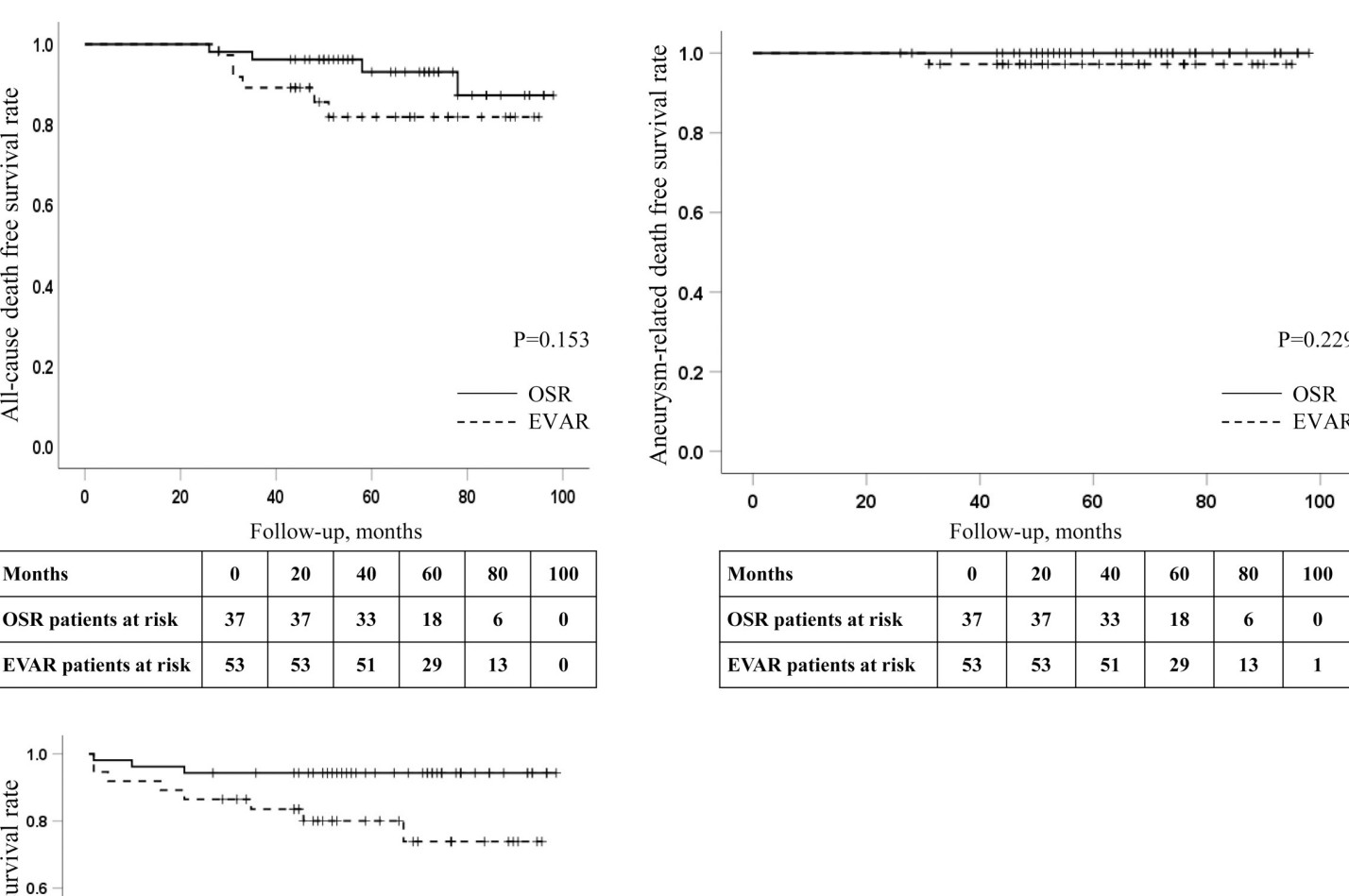

**Fig 2. Survival curves.** EVAR, Endovascular aneurysm repair; OSR, open surgical repair.

EVAR (mean, $1307; range, $1295–$1727) vs. OSR (mean, 196; range, $0–$2591) were significantly lower (both P < .001) in the EVAR group. The complication-associated costs were lower in the OSR group (mean, $188; range, $0–$5149) than those in the EVAR group (mean, $1474; range, $0–$16 864); however, the difference was not statistically significant (P = 0.087; Table 6).

**Table 3. Complication (requiring readmission) rates during follow-up.**

| No. of complications | EVAR | | OSR | | Recurrent events model[a] | |
|---|---|---|---|---|---|---|
| | No. of events | Average time to event (mns) | No. of events | Average time to event (mns) | HR (95% CI)[b] | P-value |
| **1st complication (including reintervention)** | 8 | 23.25 | 3 | 10 | .11 (0.03–0.45) | .0022 |
| **2nd reintervention** | 4 | 19.75 | | | | |
| **3rd reintervention** | 1 | 1 | | | | |

HR for OSR group (reference group = EVAR).

[a]Andersen and Gill model with robust sandwich estimator.

[b]Adjusted for age.

EVAR, Endovascular aneurysm repair; OSR, open surgical repair; mns, months; HR, hazard ratio; CI, confidence interval.

# Discussion

Since the first successful EVAR procedure in the 1990s, the application of EVAR for AAA has steadily increased [16]. This popularity is a result of the numerous studies that demonstrated desirable perioperative and short-term outcomes for EVAR compared with those for OSR [2,17–20]. However, subsequent studies on the same topic suggest little difference in the long-term outcomes between EVAR and OSR [1,4,15,17,18,21,22]. Some studies have reported better long-term outcomes for OSR [3,20,23,24], adding more contradictory data to this controversial topic. Our center also reported the superiority of OSR in long-term outcome [24]. The

**Table 4. Details of complications and medical expenses.**

| EVAR | | | | |
|---|---|---|---|---|
| No. of patients | Indication for reintervention | Reintervention | Medical expense ($) | Reintervention-free duration (mns) |
| 1 | T2EL | Embolization | 4561 | 15 |
| | T2EL (surgery) | Open graft interposition | 12 304 | 24 |
| 2 | T2EL | Embolization | 3013 | 34 |
| | T2EL | Embolization | 2725 | 58 |
| | T1bEL | Limb graft extension | 6977 | 59 |
| 3 | T1bEL | Stent graft insertion | 19 955 | 66 |
| 4 | T2EL | Embolization | 5693 | 45 |
| 5 | T1aEL | Embolization | 3315 | 4 |
| 6 | T2EL | Stent graft insertion | 5812 | 20 |
| | Endotension (surgery) | Open proximal reinforce and redo EVAR | 17 235 | 39 |
| 7 | Limb occlusion | Thrombolysis and stent graft insertion | 13 226 | 1 |
| 8 | T1aEL | Embolization | 7658 | 1 |
| | T1aEL | Aortic cuff implantation | 4704 | 28 |
| Total expense of intervention | | | 107 178 | |
| Mean expense per intervention | | | 8245 | |
| OSR | | | | |
| No. of patients | Indication for admission | Treatment | Medical expense ($) | Treatment-free duration (mns) |
| 1 | Hernia | Mesh herniorrhaphy | 5149 | 20 |
| 2 | Mechanical ileus | Conservative care | 1386 | 1 |
| 3 | Hernia | Hernia repair | 3421 | 9 |
| Total expense of complication | | | 9956 | |
| Mean expense per complication | | | 3319 | |

EVAR, Endovascular aneurysm repair; OSR, open surgical repair; T2EL, type 2 endoleak; T1bEL, type 1b endoleak; T1aEL, type 1a endoleak; mns, months.

**Table 5. Medical expenses and cost per QALY.**

|  | EVAR | OSR | P-value[†] |
|---|---|---|---|
| **In-hospital cost, $** | 16 498 (10 905–34 296) | 10 523 (7615–25 105) | < .001 |
| **Cost until 1 year, $** | 18 014 (11 769–36 132) | 10 679 (7615–25 536) | < .001 |
| **Cost until 2 years, $** | 19 140 (12 201–36 564) | 10 841 (7615–28 828) | < .001 |
| **Cost until 3 years, $** | 19 279 (12 201–36 564) | 10 907 (7615–28 828) | < .001 |
| **QALY at 3 years (95% CI)** | 2.49 (2.27–2.70) | 2.75 (2.69–2.82) | .013 |
| **Cost per QALY at 3 years, $** | 10 137 (4454–68 419) | 4038 (2705–12 545) | < .001 |

Values are reported as mean (range), except when indicated.

Costs are the accumulated values.

† Mann-Whitney *U* test.

EVAR, Endovascular aneurysm repair; OSR, open surgical repair; QALY, quality-adjusted life year; CI, confidence interval.

main reason for the EVAR long-term outcome being lower than OSR long-term outcome was the reintervention rate. The high reintervention rate was related with the high cost and QALY. Young populations who have a longer life expectancy than old populations are more exposed to chances for reintervention, leading to a higher cost and lower QALY. These young patients with AAA, who fit both OSR and EVAR, were our subject of investigation into the reintervention rate, medical expenses, and QALY. As a result, the three-year complication rate was 0.11 for HR (90% CI, 0.03–0.45; P = .0022), which was lower in the OSR than EVAR. For the three-year medical expenses, it was lower (P<0.001) in the OSR group (mean, $19 279; range; $7615–28 828) than that in the EVAR group (mean, $19 279; range, $12 200–$36 564). In addition, the three-year QALY was higher for OSR (2.75; 95% CI, 2.69–2.82) than that for EVAR (2.49; 95% CI, 2.27–2.70) (P = .013).

The patients diagnosed with AAA are at high risk of the onset of numerous comorbidities and are not expected to have average life expectancies that are comparable to those of their healthy counterparts. Thus, we need to seriously consider the HRQOL of the patients after their treatment for AAA. This study calculated the QALY at three years to assess the HRQOL. We observed higher scores in patients from the OSR group compared with those from the EVAR group, with a gain of .26 QALY at three years (Table 7). Ulug et al. reported a gain of

**Table 6. Cumulative annual cost.**

|  |  | EVAR (n = 37), $ | OSR (n = 53), $ | P-value[†] |
|---|---|---|---|---|
| *In-hospital cost* |  | 16 498 (10 905–34 296) | 10 523 (7615–25 105) | < .001 |
|  | **Follow-up cost at 1 yr** | 864 (864–864) | 65 (0–864) | < .001 |
|  | **Complication cost at 1 yr** | 652 (0–13 168) | 91 (0–3421) | .358 |
| *Cost until 1 yr* |  | 18 014 (11 769–36 132) | 10 679 (7615–25 536) | < .001 |
|  | **Follow-up cost at 2 yrs** | 1295 (1295–1295) | 130 (0–1727) | < .001 |
|  | **Complication cost at 2 yrs** | 1347 (0–16 864) | 188 (0–5149) | .087 |
| *Cost until 2 yrs* |  | 19 140 (12 201–36 564) | 10 841 (7615–28 828) | < .001 |
|  | **Follow-up cost at 3 yrs** | 1307 (1295–1727) | 196 (0–2591) | < .001 |
|  | **Complication cost at 3 yrs** | 1474 (0–16 864) | 188 (0–5149) | .087 |
| *Cost until 3 yrs* |  | 19 279 (12 201–36 564) | 10 907 (7615–28 828) | < .001 |

Values are reported as mean (range).

† Mann-Whitney *U* test.

EVAR, Endovascular aneurysm repair; OSR, open surgical repair; yr, year.

**Table 7. Primary procedure in-hospital cost (price of 2019, exchange rate of 2019).**

| | | unit cost, $ | EVAR(GA), $ | EVAR(LA), $ | OSR, $ |
|---|---|---|---|---|---|
| Preoperative workup | | | | | |
| Aortic dissection CT | | 417 | 417 | 417 | 417 |
| Radiography (chest X-ray) | | 20 | 20 | 20 | 20 |
| Echocardiography | | 290 | 290 | 290 | 290 |
| Myocardial SPECT | | 821 | 821 | 821 | 821 |
| Pulmonary function tests | | 36 | 36 | 36 | 36 |
| Laboratory | | 206 | 206 | 206 | 206 |
| Hospital stay | | | | | |
| Ward bed | | (per diem) 73 | 716* | 533** | 840*** |
| ICU bed | | (per diem) 329 | 0 | 0 | 329 |
| Anesthesia | | | | | |
| General anesthesia | | 414 | 414 | 0 | 414 |
| CVP monitor | | 59 | 59 | 0 | 59 |
| IBP monitor | | 37 | 37 | 0 | 37 |
| Image during in-hospital | | | | | |
| Aortogram | | 779 | 779 | 779 | 0 |
| Aortic dissection CT | | 417 | 417 | 417 | 417 |
| Equipment | | | | | |
| Endurant-bifurcated body | | 4024 | 4024 | 4024 | 0 |
| Endurant-contralateral limb | | 2445 | 2445 | 2445 | 0 |
| Balloon | | 429 | 429 | 429 | 0 |
| Angiographic catheter | | 34 | 34 | 34 | 0 |
| Graduated sizing catheter | | 137 | 137 | 137 | 0 |
| Lunderquist wire guide | | 57 | 57 | 57 | 0 |
| Raidofocus guide wire | | 24 | 24 | 24 | 0 |
| Introducer sheath | | 27 | 27 | 27 | 0 |
| Perclose-proglide | | 249 | 249 | 249 | 0 |
| Dacron graft (gelsoft bifurcated) | | 609 | 0 | 0 | 609 |
| Dacron graft (hemagard) | | 628 | 0 | 0 | 0 |
| Procedure cost | EVAR (GA) | 1284 | 1284 | 0 | 0 |
| | EVAR (LA) | 1254 | 0 | 1254 | 0 |
| | OSR | 1387 | 0 | 0 | 1387 |
| Overall in-hospital cost | | | 12478 | 12199 | 5467 |

* 73 (unit cost per diem) × 9.8 (mean units, days).

**73 (unit cost per diem) × 7.3 (mean units, days).

***73 (unit cost per diem) × 11.5 (mean units, days).

Unit cost is based on Asan Medical Center.

EVAR, Endovascular aneurysm repair; OSR, open surgical repair; CT, computed tomography; SPECT, single-photon emission computed tomography; ICU, intensive care unit; CVP, central vein pressure; IBP, invasive blood pressure; GA, general anesthesia; LA, local anesthesia.

.166 QALY at three years in patients with ruptured AAA (95% CI, 0.002–0.331), while there were differences in outcomes between ruptured and elective AAA [15]. Prinssen et al. reported a significant increase in the EQ-5D score at 6 months post-operation for OSR compared with that for EVAR, which is comparable to the results of our study [13].

Not only did OSR result in higher QALYs but also the total sum of costs for OSR at three years was much lower than those for EVAR. The reasons for the lower costs with OSR are as follows: 1) Lower in-hospital cost of the Korea health insurance system; 2) not having to

constantly check for endoleaks using CT (as in the case for EVAR); and 3) no costs related to reinterventions. In our study, the HR for complication was 0.11 (reference group = EVAR; P = .0022), consistent with the results of previous RCTs [1,15,25,26]. In addition to the higher total sum of costs and repeated CT procedures for EVAR, the cumulative dose of radiation administered to patients requiring reintervention must also be considered when choosing the operational procedure. OSR may therefore be more suitable for younger patients.

In this study, there was no perioperative mortality in both the EVAR and OSR groups. Both EVAR and OSR had to be studied in young patients who were of the highest similarity as possible; therefore, patients with a life expectancy of less than one year, as well those with unsuitable anatomical morphology, were excluded. This is the most likely reason that no perioperative mortality was observed during this study. Of the 59 patients under the age of 70 years who underwent EVAR during the study period, 14 were excluded owing to their life expectancy of less than one year. Among the 118 patients under the age of 70 years who underwent OSR, no patients had a life expectancy of less than one year. According to this study on patients treated with AAA between 2001 and 2012 at our center, 7 of 352 patients (1.9%) with non-ruptured AAA died in the hospital. The two-,five-, and ten-year survival rates were 94.6%, 89.9%, and 83.4%, respectively [27]. In a study on patients over 40 years of age treated with AAA between 2014 and 2016, the 30-day mortality was 0.8%. This study also discovered that morbidity (including renal complications) and mortality due to OSR were not higher in the suprarenal aortic clamping group than those in the infrarenal aortic clamping group [28].

The mean total lengths of hospital stay were 6.3 and 9.7 days for EVAR and OSR, respectively. This result for EVAR was similar to those of four RCTs (EVAR 1 trial, 8.34 days; DREAM, 6 days; ACE trial, 5.8 days; and OVER, 5 days) [14]. The result for OSR was also similar to that of the ACE trial (8 days) [17]. Previous reports have suggested that EVAR results in lower costs than OSR due to shorter hospital stays in EVAR patients that offset the costly medical bills. The studies also reported lower 30-day mortality and morbidity rates for EVAR than those for OSR, which may also explain the lower costs for EVAR [29]. However, our results indicated a shorter length of in-hospital care for EVAR than that for OSR, and little difference in the 30-day mortality and morbidity rates between them. Despite this, the total sum of costs was lower for OSR. According to Epstein et al. [30], EVAR had long-term cost effectiveness that was comparable to that for OSR in the OVER trial conducted in the US, but was not cost effective in the trials conducted at a European center. Thus, the cost effectiveness in Korea must be analyzed based on the medical circumstances there. In Korean medical circumstances, the predominant factor associated with the lower total sum of costs for OSR was the cost of the procedures themselves. The operation-related fee in Korea is lower than that in Western countries, while the costs of endovascular devices are higher in Korea. The Endurant-bifurcated body costed 4.024 USD and the Endurant-contralateral limb costed 2,445 USD in 2019. Moreover, disposables such as balloons, catheters, and guidewires are expensive. Table 7 shows the costs of devices, examinations, and procedures according to the cost in 2019 (exchange rate, 2019 [11]). Primary procedure in-hospital costs of EVAR are much higher than those of OSR (EVAR under GA, $12 478; EVAR under LA, $12 199; OSR, $5467).

This study had several limitations. This retrospective study was conducted in a single center. The patients were able to choose the operation method and procedure; thus, the study was not randomized. Furthermore, although vascular surgeons and radiologists discussed with each other, it is possible that different specialists performed the same procedure may affect the results. However, the study has several strengths, including the fact that we directly collected telephone survey responses for the EQ-5D score after the procedure. We used the average score during the first three years after the primary procedure. This could be a factor for enabling us to measure the QALYs accurately.

In conclusion, the results of this study suggest that the complication rate is lower for OSR, and that OSR shows better cost effectiveness than EVAR in young patients, under the age of 70 years, with a suitable aortic anatomy. Therefore, OSR can be the first choice for surgeons to treat young patients with AAA and appropriate anatomical features.

## Supporting information

**S1 Table. STROBE checklist.**
(DOCX)

**S1 File. Studys minimal underlying data.**
(XLSX)

## Author Contributions

**Conceptualization:** Yong Pil Cho, Youngjin Han.

**Data curation:** Eunae Byun.

**Investigation:** Eunae Byun, Gi Young Ko, Min-Jae Jeong.

**Supervision:** Hyangkyoung Kim.

**Writing – original draft:** Eunae Byun.

**Writing – review & editing:** Eunae Byun, Tae-Won Kwon, Hyangkyoung Kim.

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
