## [Decision Letter · Decision Letter 0]

12 Aug 2021

PONE-D-21-24097

Comparison of the quality of life and cost-effectiveness of endovascular versus open surgical repair for abdominal aortic aneurysm in young patients

PLOS ONE

Dear Dr. Kwon,

Thank you for submitting your manuscript to PLOS ONE. After careful consideration, we feel that it has merit but does not fully meet PLOS ONE’s publication criteria as it currently stands. Therefore, we invite you to submit a revised version of the manuscript that addresses the points raised during the review process.

Please revise accordingly.

We look forward to receiving your revised manuscript.

Kind regards,

Academic Editor

PLOS ONE

Journal Requirements:

Reviewers' comments:

Reviewer's Responses to Questions

**Comments to the Author**

1. Is the manuscript technically sound, and do the data support the conclusions?

Reviewer #1: Partly

Reviewer #2: Partly

2. Has the statistical analysis been performed appropriately and rigorously? 

Reviewer #1: Yes

Reviewer #2: Yes

3. Have the authors made all data underlying the findings in their manuscript fully available?

Reviewer #1: Yes

Reviewer #2: Yes

4. Is the manuscript presented in an intelligible fashion and written in standard English?

Reviewer #1: Yes

Reviewer #2: Yes

5. Review Comments to the Author

Reviewer #1: First of all, congratulate the authors for the idea and the work presented. There are some controversial aspects about the advantages of one technique or another in certain groups of patients and the results of the treatment in a short, medium and long term.

Next, a series of recommendations are made to the authors for their consideration.

Why did they choose that period of time for patient inclusion and on which they based the period of time necessary for patient follow-up?

Line 165 statistical analysis It is important that the authors indicate how the sample size was calculated or, failing that, with the analyzed sample, how we can be sure that the data is statistically significant and relevant?

Authors should explain what the operator's experience was, and what it depended on being performed by a radiologist or vascular surgeon.

line 67 please include the reference.

Please define what you understand by high anesthetic risk and how you estimated the risk of each patient.

It would be important for the authors to include the cause of the aneurysm (aetiology), that is, if it is of atherosclerotic origin related to hypertension or is it due to an arteriopathy.

Were any criteria used for the EVAR to be performed by radiologists or vascular surgeons?

Line 136 outcomes. It would be very interesting to know data from both groups such as mortality, complications in the immediate postoperative period or the length of hospital stay.

Line 155 the authors do not have to describe what the EQ-5D is. It would be important that the authors indicate how often and when the questionnaire was passed to the patients. It is to be expected that at some moments of the follow-up there would be differences between the groups.

Table 3 is misleading in interpretation. When the 1st complication including reoperation is indicated, really in the OSR group no patient required reoperation due to the pathology. Redo the statistical analysis removing these three patients from the study.

Table 4 also leads to false interpretation. The average cost of the process is indicated. Include the total cost of complications in each of the groups.

Table 5 please review the data of in-hospital cost EVAR since the cost reported is not included in the cost range

The same happens in table 6.

In the discussion section, the authors include data that are part of the results, for example the mean stays or the long-term survival rate. Please, review it.

The complications that appear in the limitations must be transferred to the results, furthermore this is not a limitation, if it could be the sample size but not the appearance of these complications.

Consider including in the limitations the form of inclusion of patients in each of the two branches of the study. In addition, it can also be a limitation not knowing what caused the same procedure to be performed on some occasions by a radiologist and on others by a surgeon. Another limitation of the study is the cost analysis, the authors do not individually consider the cost of staff (surgeons, radiologists, nurses)

PRISMA checklist should be added as supplemental table.

Please, review reference 1.

Reviewer #2: - Purpose: To evaluate quality of life and cost-effectiveness up to a 3-years FU between EVAR and OSR in patients under 70-y-o with a surgical aneurysm of abdominal aorta. The authors base their analysis on QALY which stands for “Quality Adjusted Life Year”. The QALY is commonly used in health economic evaluations as a means of quantifying the health effect of a medical intervention or a prevention program and ultimately to help payers allocate healthcare resources. It is however rather imprecise to determine the true QOL for which specific scales are proposed (but not used in the present study). I therefore recommend that the title of the manuscript would be changed accordingly: “Quality Adjusted Life Year comparison at medium term follow-up….”

- Methods: Retrospective study including 90 patients <70-y-o for whom aortic morphology would have allowed to perform TEVAR and who underwent TEVAR (37) or OSR (53) between January 2012 and October 2016.

• It is well known that studies which retrospectively screen if certain patients would have been eligible for one interventional treatment rather than for another one are flawed by many biases. Among those biases are the technological evolution of imaging, materials, and indications as well as the improvement of the individual experience and global quality of care. Those biases tend to render rather futile any comparison of patient populations treated at different time and to invalid in a more recent patient population some general results found in an historical study. For instance to compare patients who, very restrictively, benefited from TAVR between 2012 and 2016 which those who are liberally treated nowadays with TAVR updated prostheses does not make much sense. Even worse if we consider studying patients who “might have benefited from TAVR between 2012 and 2016”. To deal with such biases in the current study the authors chose to select only patients who met the nowadays aortic morphology criteria for EVAR and retrospectively reviewed all preoperative CT.

- It is important to precise who is behind the “we analysed” preoperative CT. Was it the same observer for all CT? Were results double checked? Was a certified radiologist present among authors? Which imaging software was used?

- It is crucial therefore that the authors make it clear in “Methods” that not a single patient from the (historical) EVAR group did not meet the nowadays aortic morphology criteria for EVAR because this might significantly increase the risk of impaired outcomes during follow-up.

• On the other hand, to avoid including patients that might have not tolerate OSR, the authors have excluded 14 patients for EVAR who would not have tolerate general anaesthesia. However the EVAR group did include 18 procedures (50%) performed under local anaesthesia. The authors must therefore precise the criteria of choice for local versus general anaesthesia in the EVAR group and how they can retrospectively be sure that patients who underwent EVAR under LA could have tolerate GA as well?

- Results: Even in case of a favourable anatomy, a lesser invasive treatment tend to be proposed in frail or disable patients. Patients with a past history of cancer are also good candidate for a lesser invasive treatment. The authors present a baseline comparison of demographics (Table 1) which is in line with this general figure. Due to the low number of patients and to assess a better comparison between both groups (taking into account that this is not a matched study), I strongly recommend that the authors present in Table 1 the Charlson Comorbidity Index for EAVR and OSR patients.

• A higher rate of complications is expected with EVAR during follow-up and patients must stay therefore under imaging surveillance for a life time which impacts QOL. Some large recent randomised trials share those unequivocal conclusions, however a lower global survival with EVAR in the long term is not unanimously found (please do actualise bibliography, for instance we expect to find DOI: 10.1056/NEJMoa1715955). I strongly recommend to present actuarial survival curves (freedom from death of any cause, freedom of death from aortic cause, and freedom of reintervention) in a Figure 1.

• HRQOL (Health Related Quality of Life) is usually assessed by a score on a HRQOL-14 items scale. Since such a scale has not been used, I recommend the authors to strictly stay on the notions of QALY which can be assimilated to the crude survival in complete heath for each group and cost per QALY which are relevant in this study up to a medium term FU.

• Finally minor English revisions are required, please check carefully manuscript (for instance: “Heria repair” in Table 4; “Fallow-up cost” in Table 6; “follow-ups” in Cost analysis…..)

6. PLOS authors have the option to publish the peer review history of their article (what does this mean?). If published, this will include your full peer review and any attached files.

Reviewer #1: No

Reviewer #2: **Yes: **Pr. Dr. CAUS Thierry, MD, PhD

---

## [Author Response · Author response to Decision Letter 0]

28 Oct 2021

Reviewer 1: We have incorporated all of your suggestions into my revision. They were very helpful. We uploaded our response as an attachment. Thank you for your help.

Reviewer 2: We have incorporated all of your suggestions into my revision. They were very helpful. We uploaded our response as an attachment. Thank you for your help.

---

## [Decision Letter · Decision Letter 1]

16 Nov 2021

Quality Adjusted Life Year comparison at medium term follow-up of endovascular versus open surgical repair for abdominal aortic aneurysm in young patients

PONE-D-21-24097R1

Dear Dr. Kwon,

We’re pleased to inform you that your manuscript has been judged scientifically suitable for publication and will be formally accepted for publication once it meets all outstanding technical requirements.

Kind regards,

Academic Editor

PLOS ONE

Additional Editor Comments (optional):

Reviewers' comments:

Reviewer's Responses to Questions

**Comments to the Author**

1. If the authors have adequately addressed your comments raised in a previous round of review and you feel that this manuscript is now acceptable for publication, you may indicate that here to bypass the “Comments to the Author” section, enter your conflict of interest statement in the “Confidential to Editor” section, and submit your "Accept" recommendation.

Reviewer #1: All comments have been addressed

Reviewer #3: All comments have been addressed

2. Is the manuscript technically sound, and do the data support the conclusions?

Reviewer #1: Yes

Reviewer #3: Yes

3. Has the statistical analysis been performed appropriately and rigorously? 

Reviewer #1: Yes

Reviewer #3: Yes

4. Have the authors made all data underlying the findings in their manuscript fully available?

Reviewer #1: Yes

Reviewer #3: Yes

5. Is the manuscript presented in an intelligible fashion and written in standard English?

Reviewer #1: Yes

Reviewer #3: Yes

6. Review Comments to the Author

Reviewer #1: Congratulate the authors for the revision work they have done.

I think the final version has improved a lot.

Reviewer #3: (No Response)

7. PLOS authors have the option to publish the peer review history of their article (what does this mean?). If published, this will include your full peer review and any attached files.

Reviewer #1: **Yes: **Juan Bustamante-Munguira MD PhD MsC MPH Head Cardiac Surgery Department. University Hospital of Valladolid. Spain.

Reviewer #3: No

---

## [Editor Report · Acceptance letter]

23 Nov 2021

PONE-D-21-24097R1 

Quality Adjusted Life Year comparison at medium term follow-up of endovascular versus open surgical repair for abdominal aortic aneurysm in young patients 

Dear Dr. Kwon:

I'm pleased to inform you that your manuscript has been deemed suitable for publication in PLOS ONE. Congratulations! Your manuscript is now with our production department. 

Kind regards, 

on behalf of

Dr. Robert Jeenchen Chen 

Academic Editor

PLOS ONE